# Thermal Properties of Porous Silicon Nanomaterials

**DOI:** 10.3390/ma15238678

**Published:** 2022-12-05

**Authors:** Aleksandr S. Fedorov, Anastasiia S. Teplinskaia

**Affiliations:** 1International Research Center of Spectroscopy and Quantum Chemistry, Siberian Federal University, 660041 Krasnoyarsk, Russia; 2Kirensky Institute of Physics, Federal Research Center KSC SB RAS, 660036 Krasnoyarsk, Russia

**Keywords:** porous silicon, aerogel, thermal properties, heat capacity, molecular dynamics

## Abstract

The thermal properties, including the heat capacity, thermal conductivity, effusivity, diffusivity, and phonon density of states of silicon-based nanomaterials are analyzed using a molecular dynamics calculation. These quantities are calculated in more detail for bulk silicon, porous silicon, and a silicon aerocrystal (aerogel), including the passivation of the porous internal surfaces with hydrogen, hydroxide, and oxygen ions. It is found that the heat capacity of these materials increases monotonically by up to 30% with an increase in the area of the porous inner surface and upon its passivation with these ions. This phenomenon is explained by a shift of the phonon density of states of the materials under study to the low-frequency region. In addition, it is shown that the thermal conductivity of the investigated materials depends on the degree of their porosity and can be changed significantly upon the passivation of their inner surface with different ions. It is demonstrated that, in the various simulated types of porous silicon, the thermal conductivity changes by 1–2 orders of magnitude compared with the value for bulk silicon. At the same time, it is found that the nature of the passivation of the internal nanosilicon surfaces affects the thermal conductivity. For example, the passivation of the surfaces with hydrogen does not significantly change this parameter, whereas a passivation with oxygen ions reduces it by a factor of two on average, and passivation with hydroxyl ions increases the thermal conductivity by a factor of 2–3. Similar trends are observed for the thermal effusivities and diffusivities of all the types of nanoporous silicon under passivation, but, in that case, the changes are weaker (by a factor of 1.5–2). The ways of tuning the thermal properties of the new nanostructured materials are outlined, which is important for their application.

## 1. Introduction

Nanoporous silicon is a solid material with a porous structure consisting of pores smaller than 100 nm. Owing to the unique flexibility of the physicochemical properties (porosity, pore size, passivation of the internal surface with different substances, etc.) of porous silicon, it has become increasingly interesting for both fundamental research and application.

Modifying the inner surface of porous silicon, one can change its optical properties [1], which makes it promising for application in chemical and biochemical sensors. Another remarkable example of the modification is filling porous silicon with drugs [2]. In [3], it was shown that nanocrystal silicon functionalized with triethoxyvinylsilane exhibited a structure-size-dependent photoluminescence, which is important for optoelectronics.

In addition, it was shown that porous silicon could be used in the biodetection of miRNA on the basis of the surface plasmon resonance and surface-enhanced Raman spectroscopy (SERS) [4].

Tuning the degree of porosity and pore size in porous silicon, one can improve the performance of fuel cells which convert chemical energy into electric energy and, consequently, improve the light energy conversion [5].

The ways of enhancing the energy storage density of porous-silicon-based lithium-ion batteries [6] and silicon-nanographite aerogel [7] using the flexibility of the porous silicon’s shape were reported. The structural integrity of porous silicon can be maintained upon intercalation of lithium into silicon, which leads to a significant lithiation-induced volumetric expansion of silicon [8]. In addition to porous silicon, other silicon nanostructures, for example, nanoparticles [9] and nanowires [10], can be used for these purposes.

Porous silicon also finds application as a wide-gap absorber for solar cells. Its advantage is its high surface area and ability to convert the high-energy ultraviolet and blue ranges of the solar spectrum [11].

Concerning different silicon nanostructures, there exist many methods for improving the energy storage with their use. For example, as has recently been shown, silicon nanohorns can improve the performance of solar-driven water-splitting devices, which ensure a better energy conversion than their bulk counterparts [12]. In [13], it was shown that large-area, ordered Si nanowire arrays can also be used for this purpose.

Silicon nanowires, porous membranes, and nanomeshes can convert heat flows directly into useful electric energy, which makes it possible to apply these structures in thermoelectric generators [14].

As for the thermal properties of nanoporous silicon, the most interesting application concerns thermoresistive sensors. Recently, it has been established that porous silicon, in contrast to the bulk material, can solve the problem of the thermal incompatibility of materials, which is met in thermal sensors [15,16]. It becomes possible to tune the desired physical properties of sensors using only porous silicon.

It was shown that porous silicon microhotplates can be used to improve thermal isolation in silicon temperature sensors due to their compatibility with a standard complementary metal–oxide–semiconductor (CMOS) technology [17].

Another promising area for the use of the thermal properties of nanosilicon is thermal energy storage (TES) [18]. To save energy as sensible heat, the following properties are important: a high heat storage capacity, a good thermal conductivity for easy charging–discharging, stability over a great number of operation cycles, and a high melting point [19]. Since porous silicon satisfies these conditions, it can be used for thermal energy storage.

In addition, it is noteworthy that the silicon oxide nanomaterials have a high heat capacity and are used as core-shell nanostructures for TES applications [20].

In recent study [21], the heat capacity and thermal conductivity of porous silicon thin films were experimentally examined. The monotonic growth of the heat capacity as a function of the degree of porosity was first established. In addition, the porosity dependence of the thermal conductivity for the initial and passivated porous silicon films were obtained.

In this study, a theoretical approach to determining the heat capacity, thermal conductivity, and phonon density of states (PDOS) of silicon-based nanomaterials (porous silicon and silicon aerocrystal) is first proposed. The investigations are carried out by a classical molecular dynamics (MD) method.

## 2. Thermal Properties of Porous Silicon

By now, it has been found that nanomaterials have a high heat capacity, which was demonstrated, in particular, for gold [22], titanium oxide [23], silicon oxide [24], lead sulfide [25], and copper oxide [26]. Here, we consider the factors affecting the thermal properties of porous silicon.

First of all, it should be emphasized that the contribution of the electron gas to the heat capacity and thermal conductivity around room temperature can be ignored because of the dielectric behavior [27]. We assume that the thermal properties of the material under study depend only on phonon lattice vibrations, whose energy is (Equation 1)
(1)$$E\;=\;\sum_{q,m}\frac{\hbar \omega_{q,m}}{\exp(\frac{\hbar \omega_{q,m}}{kT})\;-\;1}$$
where $\omega_{q,m}$ is the angular frequency of the phonon mode as a function of the wave vector **q** and the oscillation mode *m*.

Previously [26], a model for describing the heat capacity of nanoparticles was proposed in which surface atoms and atoms inside a nanoparticle were considered separately. Here, this approach is used to find the heat capacity of a porous material, since both silicon nanoparticle and porous silicon are nanosized and have an empty space with the porosity $P\;=\;\frac{\rho_{bulk}\;-\;\rho_{porous}}{\rho_{bulk}}$, where $\rho_{bulk}$ is the density of the bulk material and $\rho_{porous}$ is the density of the porous material containing voids.

The heat capacity of porous silicon can be determined by differentiating (Equation 1). The constant-pressure heat capacity can be expressed as a function of frequency:$\omega_{q,m}$:(2)$$C_{P}\;=\;\frac{1}{kT^{2}}\sum_{q,m}\frac{{\left(\hbar \omega_{q,m}\right)}^{2}\exp\left(\frac{\hbar \omega_{q,m}}{kT}\right)}{{\left(\exp\left(\frac{\hbar \omega_{q,m}}{kT}\right)\;-\;1\right)}^{2}}$$

The propagation of phonons in nanomaterials differs strongly from the case of bulk materials [25]. In the cited work, it was demonstrated what happens with atomic vibrations when nanoparticles have a size within the meso- (2–50 nm) or microscale (<2 nm) limit and, consequently, an internal surface and a great number of surface atoms with changed bonds. The surface atoms make a great contribution to the atomic amplitudes and the total energy of different oscillation modes, which leads to a PDOS shift g(ω) to the low-frequency spectral region (the softening effect). When the ratio between the number of surface atoms and the total number of atoms is high, this effect should be taken into account.

This PDOS shift was demonstrated using the MD simulation for silicon nanoparticles [28]. However, in the cited work, the silicon nanoparticle surface was not passivated. In addition, it was found that the smaller the silicon nanoparticle size, the stronger the heat capacity growth. Importantly, the high heat capacity cannot be observed at a particle size larger than 50 nm [29] due to the relatively small internal surface area.

The heat capacity was determined using the approach that splits the phonon contribution into the Einstein model for optical phonons and the Debye model for acoustic phonons. Therefore, the total heat capacity of porous silicon was determined as a sum of the optical and acoustic heat capacities [26] $C_{P}\;=\;C_{P,O}\;+\;C_{P,A}$.

Within the Einstein model, the average oscillation frequency $\bar{\omega }$ of optical phonons can be determined by the weighted summation of the phonon oscillations of the surface and internal atoms as $\bar{\omega }\;=\;xL\omega \;+\;\left(1\;+\;x\right)\omega .$, where $x\;=\;\frac{N^{S}}{N}$ is the ratio between the number of atoms on the surface and the total number of atoms, $L\;=\;\sqrt{\frac{Z^{S}}{Z}}$ is the softening factor, $Z^{S}$ is the average number of atomic bonds on the surface, and *Z* is the average number of atomic bonds in the crystal.

Thus, the contribution of 3N optical phonons to the specific heat capacity should be determined as
(3)$$C_{P,O}\;=\;\frac{3N}{kT^{2}}\frac{\left(\hbar \bar{\omega }\right)\exp\left(\frac{\hbar \bar{\omega }}{kT}\right)}{{\left(\exp\left(\frac{\hbar \bar{\omega }}{kT}\right)\;-\;1\right)}^{2}}.$$

The phenomenon of the oscillation frequency lowering is observed also for acoustic phonons. According to the Debye assumption, the effective speed of sound for surface atoms is $v^{S}\;=\;Lv$; therefore, the contribution of acoustic phonons to the specific heat should be
(4)$$\begin{array}{c} C_{P,A}\;=\;\frac{1}{kT^{2}}(\sum_{q\subseteq Q^{I}}\frac{{\left(\hbar \omega_{q}\right)}^{2}\exp\left(\frac{\hbar \omega_{q}}{kT}\right)}{{\left(\exp\left(\frac{\hbar L\omega_{q}}{kT}\right)\;-\;1\right)}^{2}}\;+\;\sum_{q\subseteq Q^{S}}\frac{{\left(\hbar L\omega_{q}\right)}^{2}\exp\left(\frac{\hbar L\omega_{q}}{kT}\right)}{{\left(\exp\left(\frac{\hbar L\omega_{q}}{kT}\right)\;-\;1\right)}^{2}}) \end{array}$$
where $Q^{I}$ is the set of wave vectors of internal atoms and $Q^{S}$ is the set of wave vectors of surface ones. Thus, the heat capacity of porous silicon can be determined by summing Formulas (Equation 3) and (Equation 4).

The presence of internal voids in porous silicon also affects the thermal conductivity and can be described within the following model. Porous silicon consists of a solid phase, and voids and only the solid phase is assumed to be conducting. The thermal conductivity $\kappa_{PS}$ of porous silicon depends on both the porosity and distribution of solid silicon over the porous system. This can be described by the formula [30] $\kappa_{PS}\;=\;fg_{0}\kappa_{S},$ where f = (1 − P) is the volume fraction, *P* is the porosity, $g_{0}$ is the percolation strength depending on the microtopology of the porous matrix, and $\kappa_{S}$ is the thermal conductivity of solid silicon. The percolation strength $g_{0}$ can be understood in the Looyenga effective medium model [31], in which the $g_{0}$ value is related to the volume fraction *f* of the solid phase as $g_{0}\;=\;f^{2}$.

It is well-known that the thermal conductivity $\kappa_{S}$ of bulk silicon, according to the phenomenological phonon diffusion model [32], can be defined as $\kappa_{S}\;=\;\frac{1}{3}\rho c_{\nu }\nu \lambda$, where ρ, $c_{\nu }$, ν, and λ are the density, mass-specific heat capacity, speed of sound, and mean free path of phonons in silicon, respectively. The phonon’s mean free path λ can be substituted by the mean size $d_{k}$ of silicon crystallites.

Thus, the thermal conductivity of porous silicon is
(5)$$\kappa_{PS}\;\approx \;\frac{1}{3}f^{3}\rho c_{\nu }\nu d_{k}.$$

## 3. Models of Porous Silicon

Porous silicon is a solid containing free space in its volume. According to experimental data, this can be channels or pores of different shapes [19] or diverse interconnected columns, which correspond to the silicon-based aerogel or aerocrystal. An aerocrystal is porous silicon with a porosity of 90–99%.

The three porous silicon cells prepared for the investigations were: (I) an aerocrystal (Model 1, M1) with a width of 30 Å, (II) a round pore (Model 2, M2) with a width of 26 Å, and (III) a square pore (Model 3, M3) with a width of 26 Å, see Figure 1. For all three cells, the periodic boundary conditions were established. The prepared structures were consistent with the experimentally observed geometry of porous silicon [19] and silicon aerocrystals [33]. The data on the general properties of these three models are given in Table 1 with disregard for the internal surface passivation.

Along with the porous silicon models, a bulk silicon supercell with 12 × 12 × 12 cubic cells was created. Figure 2 shows the radial distribution function (RDF) for bulk silicon with the experimental lattice parameters and the RDF after relaxation of the silicon cell at a temperature of *T* = 300 K.

Porous silicon was obtained by electrochemical etching, during which a part of silicon evaporated in the form of a combination with –OH or –F and left pores. When the void formation stage is completed, one of the various methods for drying and modifying the inner surface should be used; otherwise, silicon atoms will continue to separate [19]. Therefore, the inner surface of the experimental samples was subjected to passivation. The numerous spectroscopy data and the data obtained by the drying method were presented in [19].

In the models built, the surface silicon atoms had dangling bonds; therefore, it was necessary to fill the surface with appropriate substances, specifically, the –OH, –H, and –O ions. A passivating ion was added to each silicon atom that had no filled bonds.

In practice, there are different ways of controlling the porosity, pore size, and internal structure of pores to obtain a material with desired properties. These are, in particular, optical interferometry, photoacoustics [34], and impedancemetry [35]. Different pore types have already been described: blind, interconnected, completely isolated, and through. Different pore shapes can also be obtained, for example, cylindrical, inkwell, funnel, cuboid, triangular, pyramidal, etc.

## 4. Molecular Dynamics Calculations

The classical MD calculation was performed with the large-scale atomic/molecular massively parallel simulator (LAMMPS) [36] using the reactive force field (ReaxFF) potential [37] with the KOKKOS accelerator for systems with a great number of atoms [38]. The ReaxFF was chosen because it is flexible and computationally efficient and makes it possible to describe all formations between selected groups of atoms and even chemical reactions [39]. The ReaxFF potential allowed us to carry out the MD simulation of silicon and porous silicon with the internal surface passivated with the –OH, –O, and –H chemicals groups. The RDF of the silicon cell obtained in the course of the MD simulation is presented in Figure 2.

**Figure 2 materials-15-08678-f002:**
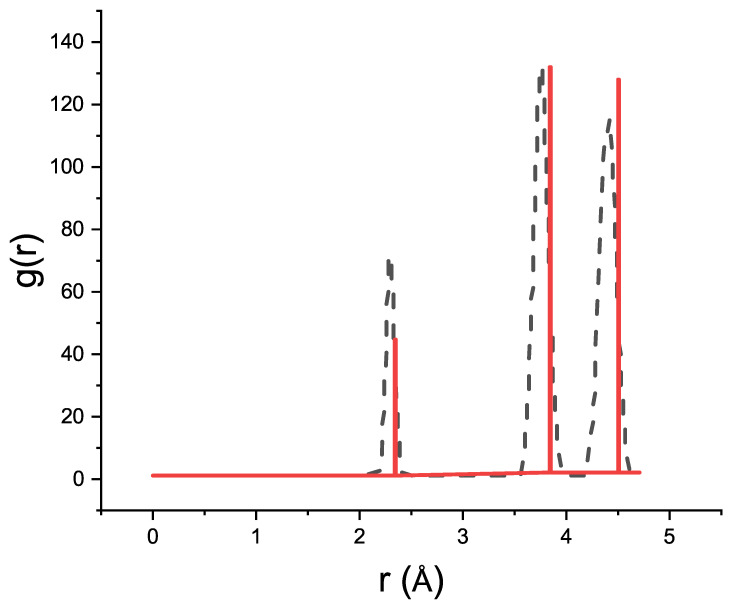
RDF of bulk silicon. The black dashed line corresponds to the RDF of silicon during the MD simulation at a temperature of 300 K and the red solid line to the RDF of bulk silicon with the experimental lattice parameter.

Before starting the MD calculation, it was necessary to optimize the atomic coordinates and minimize the total energy of the supercell; otherwise, it would have been impossible to carry out a reliable calculation for a required time interval. Upon relaxation, the total energy of the system decreased within the first 150–500 ps of the simulation and then achieved a plateau. The constancy of the total energy at a certain period of time was chosen as a criterion of the system relaxation. All the calculations were carried out at a temperature of 300 K.

When the inner surface was coated with hydroxide ions, there was a single difficulty that occurred during the relaxation of the simulated system. The total energy values showed the absence of convergence to a constant value or a plateau for the MD simulation time. Therefore, it was decided to relax the system at an elevated temperature (600 K) until it reached a constant energy.

When there are hydrogen atoms on the passivated surface, it is important to choose an appropriate MD simulation time step. In our case, a step of 0.2 fs or less should be chosen, since hydrogen has a relatively low weight [40]. If the time step were larger, then the structure would destruct itself, and the energy would sharply increase. Therefore, in calculating the heat capacity and thermal conductivity, a step of 0.2 fs was used.

Only when the porous silicon system was appropriately relaxed, did it become possible to implement the MD run for calculating the heat capacity, thermal conductivity, and PDOS.

The heat capacity can be calculated by the direct and indirect methods. The indirect method [41] is based on Formula (Equation 6) for the canonical ensemble with a constant number of particles, constant volume, and constant temperature.
(6)$$C_{V}\;=\;\frac{{\left(\bar{E^{2}}\;-\;{\bar{E}}^{2}\right)}_{NVT}}{kT^{2}}$$

Here, the calculation was carried out through the total energy fluctuations for a certain period of time. The heat capacity data should be released every 1 ps, because the energy fluctuation period for the bulk silicon is about 0.5 ps. To increase the calculation accuracy, it is necessary to run through more fluctuations; therefore, the calculation should be carried out for more than 500 ps, which makes it time consuming.

In contrast to (Equation 6), the direct method [42] makes it possible to reduce the calculation time. It performs the calculation according to the heat capacity definition:(7)$$C\;=\;\frac{dQ}{dT}$$
where dQ determines the amount of heat supplied to a sample upon its heating or cooling by dT degrees, i.e., gradually increasing and decreasing the cell temperature using an isothermal–isobaric ensemble under zero pressure. A high calculation accuracy could be obtained if the system was heated at a rate of 1 K/ps from 290 to 310 K and conversely with a simulation step of 0.2 fs. The data were recorded every 500 steps.

The effect of porosity on the heat capacity can be characterized by a change in the total PDOS g(ω). There are two methods for calculating the PDOS in the MD simulation, through the velocity autocorrelation function [43] and through the dynamical matrix [44]. The first method is easier to implement; it uses the discrete Fourier transform $g_{\alpha }\left(\omega \right)$ of the time-dependent velocity autocorrelation function $Z_{\alpha }\left(t\right)$, according to [43]: (8)$$\begin{array}{ccc} g(\omega ) & \;=\; & \sum_{\alpha }g_{\alpha }\left(\omega \right)\\ g_{\alpha }\left(\omega \right) & \;=\; & \int_{0}^{\tau }Z_{\alpha }\left(t\right)\exp\left(i\omega t\right)\;dt\\ Z_{\alpha }\left(t\right) & \;=\; & \frac{\langle \sum_{i}^{N}\nu_{i\alpha }\left(t\right)\nu_{i\alpha }\left(0\right)\rangle }{\langle \sum_{i}^{N}\nu_{i\alpha }^{2}\left(0\right)\rangle } \end{array}$$
where *N* is the number of atoms of any kind α in the scalar product of atoms velocities. In the calculation with the use of (Equation 8), the total simulation time was τ =  0.5 ps and the time interval dt was taken to be 0.1 fs.

Having calculated the PDOS g(ω) spectrum, we can find the heat capacity $C_{P}$ of the unit cell using the formula:(9)$$C_{P}\;=\;\frac{3N}{kT^{2}}\int_{0}^{\infty }\frac{{\left(\hbar_{\omega }\right)}^{2}\exp\left(\frac{\hbar_{\omega }}{kT}\right)}{{\left(\exp\left(\frac{\hbar_{\omega }}{kT}\right)\;-\;1\right)}^{2}}g\left(\omega \right)d\omega$$

The thermal conductivity can be determined by the Green–Kubo method [45] according to (Equation 10).

In this method, the structure thermal conductivity κ of the structure is found from the spontaneous heat current J(t) in the microcanonical ensemble simulation:(10)$$\begin{array}{ccc} \kappa & \;=\; & \frac{1}{VkT^{2}}\int_{0}^{\tau }\langle J(0)J(t)\rangle dt\\ J(t) & \;=\; & \frac{d}{dt}\sum_{i}r_{i}\left(t\right)e_{i}\left(t\right) \end{array}$$
where *V* is the cell volume, the angular brackets denote the averaging of **J** over the ensemble of the MD simulation time, and $e_{i}\left(t\right)$ is the sum of the potential and kinetic energy per atom with the time-dependent coordinate $r_{i}\left(t\right)$. The heat current autocorrelation function J(0)J(t) represents the correlation of a signal with its delayed copy of itself as a function of the delay.

When discussing the thermal conductivity, the finite-size exception should be taken into account [46]. It suggests that the phonon’s mean free path is limited by the size of a system.

The finite-size exception can be considered in two ways. The first way is to reproduce the supercell size to the bulk size. The second way is to find the correct simulation time that allows a single passage of a phonon along the supercell. In fact, if the periodic boundary conditions are established, phonons become free to travel over the cell, which leads to artificial correlations in the heat current autocorrelation function.

## 5. Discussion

In this section, the MD-simulated heat capacity, thermal conductivity, and phonon density of states for some porous silicon nanomaterials are presented.

Table 2 and Figure 3 give the heat capacity values for the three models of porous silicon calculated by the direct MD simulation method according to Formula (Equation 7). In addition, Table 2 gives the heat capacities obtained by the PDOS calculation using (Equation 9). The presented heat capacities were calculated taking into account different passivation types of the inner surface: without the inner surface or with coating the surface with the –O, –H, or –OH ions.

Generally, the calculation showed that the heat capacity $C_{P}$ of porous silicon increased with the inner surface *S*. In addition, the $C_{P}$ value depended on the type of ions filling the inner surface.

In an experimental study [21], the specific heat capacity of as-fabricated porous silicon samples ranged from 0.8 to 2.1 $\;\times \;10^{3}$ ($\frac{J}{kgK}$) at a porosity variation from 45% to 77%. Unfortunately, the pore size was not determined. It was only found that the size ranged from 4 to 70 nm. The internal surface area was neither measured nor estimated. Our MD simulation showed that the heat capacity depended on the inner surface area, as can be seen in Figure 3. One can see that the heat capacity grows monotonically by up to 30% with an increase in the inner surface of the Si nanostructures. This dependence can be explained after a thorough consideration of Formulas (Equation 3) and (Equation 4), in which the low-frequency contribution to the density of phonon states increases the heat capacity.

In addition, in contrast to experimental studies, the influence of different passivated ions could be considered in more detail in our MD simulation.

It is important to note that the porous silicon surface tends to a chemical evolution in experiments. For example, the hydrogen desorption reactions were observed in experimental porous silicon samples [47], which affected the heat capacity and thermal conductivity of the latter. In [48], an oxidation process was observed, which lasted for about an hour. In [49], the process of aging in the aqueous solutions revealed a change in the surface properties, which occurred for several days. In our MD simulation, the reactions of desorption and reabsortion of three different ions did not occur. We did not model the etching group or air, which corresponded to the experimental manufacturing process and was involved in the reactions. In addition, since the MD simulation lasted 1 ns, it was impossible to follow the effect of a relatively slow surface evolution.

Figure 4a–c present the PDOS plots calculated using Formula (Equation 8) for the three models with different filling of the inner surface. The plots demonstrate the effect of the PDOS on the heat capacity. The calculated PDOS curves reveals the effect of the PDOS shift to the low-frequency region (the softening effect). According to (Equation 2), low-frequency phonons contribute to the heat capacity of the material. As was shown in Section 2, this is due to the growth of the inner surface *S*. The larger the inner surface, the greater the low-frequency contribution to the phonon vibrations and the higher the heat capacity. Importantly, in this study, the porous silicon models were built with an inner surface ranging from 100.7 to 151.0 m2g; in practice, however, materials with a surface of 1000 m2g, see [50], were obtained; therefore, the heat capacity can be further improved via the inner surface area.

According to the data given in Table 2, to further increase the heat capacity, it is necessary to select certain passivating ions to coat the inner surface. In the MD simulation, a series of –O, –H, and –OH ions adsorbed was used, in addition to studying the effect of the inner surface.

It can be seen that the positive effect of this series on the heat capacity weakened when the –OH ions were changed for –H and –O. This can be explained by the fact that ions should have large displacement amplitudes to contribute to the PDOS shift to the low-frequency region [25]. For example, the –OH group has the largest number of local vibrational modes, which have a low frequency and therefore store the maximum thermal energy (Equation 2). As for the –O ion, its weight is much greater than that of hydrogen, so it has the maximum local oscillation frequency and the minimum influence on the heat capacity.

In addition, the thermal conductivities κ of all the structures were calculated using (Equation 10) in the MD simulation. Table 3 and Figure 5 give the κ values for the three models of bulk and porous silicon with different passivation types of the inner surface. Table 3 also gives the available experimental data [21].

To better understand the heat conduction in the investigated nanomaterials, the modified thermal conductivity $\kappa^{\prime }\;=\;\frac{S_{cell\;-\;void}}{S_{cell}}k$ was calculated for all the porous silicon nanomaterials. The parameter $\kappa^{\prime }$ redefines the thermal conductivity of the simulated supercells with the surface area $S_{cell}$ only for the area $S_{cell\;-\;void}$ filled with the material. The thermal conductivity of the surface area part not filled with the material (void) is assumed to be zero.

Using the experimental data [21] from Table 3, the parameter $\kappa^{\prime }$ was also calculated using the linear approximation of κ with respect to porosity *P*. Zero porosity corresponded to bulk silicon. However, the $k^{\prime }$ value was lower than that of bulk silicon, since the dependence of the thermal conductivity on the degree of porosity or the inner surface area is nonlinear, which can be determined using the percolation approach discussed in Section 2. It is noteworthy that $\kappa^{\prime }$ was smaller for model M3, because the crystalline skeleton became amorphous during the MD simulation.

The MD calculation data clearly showed that the thermal conductivities of all the investigated porous silicon structures depended on the porosity *P* and the kind of internal surface filling.

The result of the calculation was consistent with the previously obtained experimental data on the thermal conductivity of porous silicon [21]. Our study also showed the thermal conductivity depended on the sort of passivating ions that coated the silicon surface.

Along with the thermal conductivity, the quantities important for understanding the thermal processes that occur in a material are the thermal effusivity and thermal diffusivity, the calculated values of which are given in Table 4. The thermal effusivity or thermal responsivity of a material is determined as $\epsilon \;=\;\sqrt{\kappa \rho C_{P}}$. It helps better understand the ability of porous silicon to exchange thermal energy with the environment. The thermal diffusivity is determined as $\alpha \;=\;\frac{\kappa }{\rho C_{P}}$. It is a measure of the heat transfer rate in a material from its hot end to the cold one.

## 6. Conclusions

In this work, the thermal properties (heat capacity, phonon density of states, and thermal conductivity) of bulk silicon and three models of porous silicon nanomaterials were studied using the MD simulation. It was found that the heat capacity increased monotonically by up to 30% with an increase in the area of the porous inner surface and upon its passivation with –O, –H, and –OH ions. This increase was explained by the shift of the phonon density of states to the low-frequency spectral region caused by the large inner surface and by saving the thermal energy of local vibrational modes of the passivating ions, since the latter had large displacement amplitudes due to the softening of their chemical bonds with the surface.

It was established that the thermal conductivity of the investigated materials depended strongly on the degree of porosity and could be decreased by 1–2 orders of magnitude. It was found that the nature of the passivation of the nanosilicon inner surfaces affected the thermal conductivity of the investigated material. For example, the passivation of the surface with hydrogen did not significantly change this parameter, whereas passivation with oxygen ions reduced it by a factor of two on average and passivation with hydroxyl ions increased the thermal conductivity by a factor of 2–3.

The results of the molecular dynamics calculation were consistent with the available experimental data. The results of this study can be used to develop nanomaterials with improved thermal characteristics for various applications, including new heat storage.

## Figures and Tables

**Figure 1 materials-15-08678-f001:**
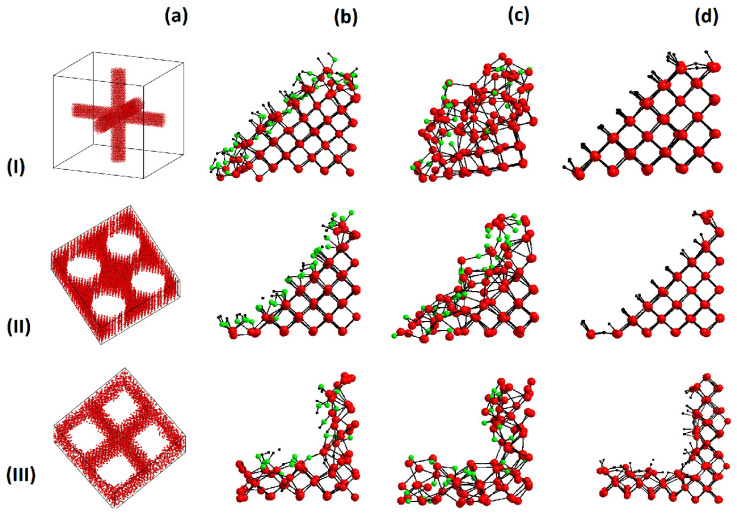
Porous silicon models built. Row I is model M1 (aerocrystal), row II is model M2 (round pore), and row III is model M3 (square pore). The columns correspond to (**a**) the initial structure, (**b**) –OH passivation, (**c**) –O passivation, and (**d**) –H passivation. Red spheres show silicon atoms, green spheres—oxygen atoms, and black spheres—hydrogen atoms.

**Figure 3 materials-15-08678-f003:**
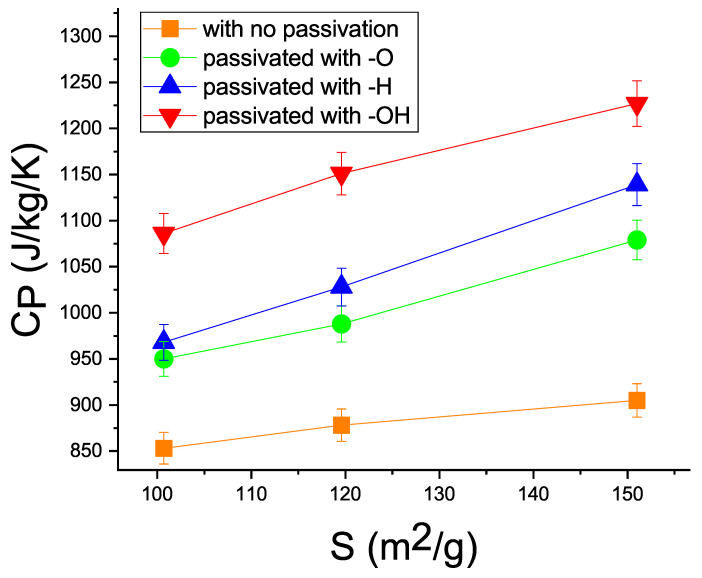
MD calculation data on the heat capacity for models M1, M2, and M3 without passivation (orange squares), at the passivation with –O (green circles), –H (blue up triangles), and –OH (red down triangles) as a function of internal surface area *S*.

**Figure 4 materials-15-08678-f004:**
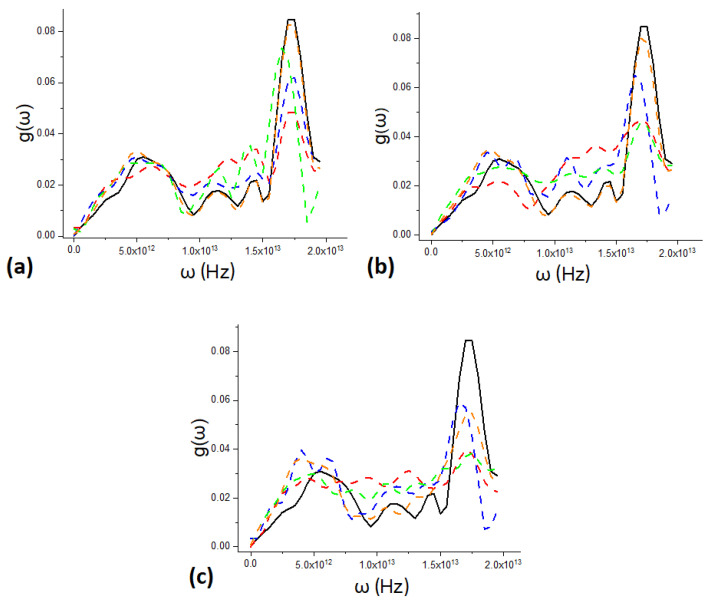
Calculated PDOS g(ω) for models (**a**) M1, (**b**) M2, and (**c**) M3. Black lines show the PDOS of bulk silicon; orange lines correspond to the absence of internal surface passivation; green lines correspond to the –O ion coating of the internal surface; blue lines, to the –H ion coating; and red lines, to the –OH ion coating.

**Figure 5 materials-15-08678-f005:**
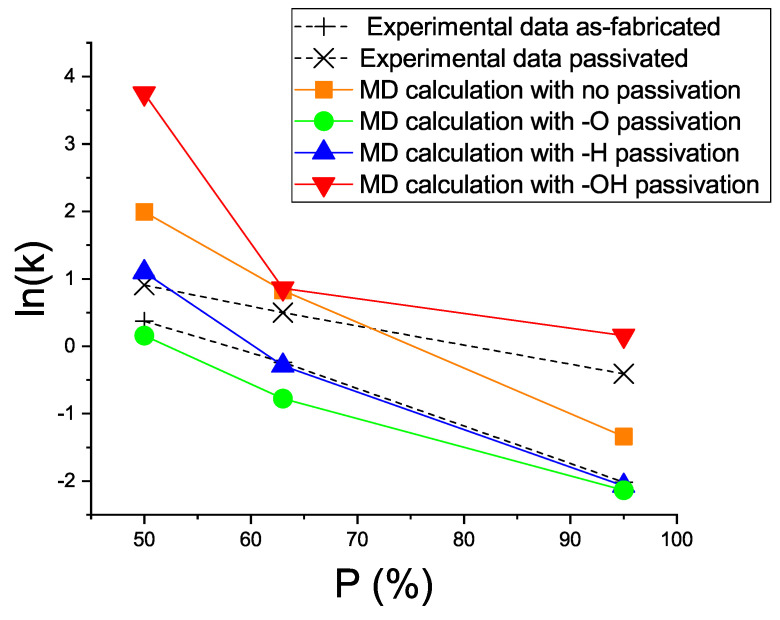
Thermal conductivity ((in logarithmic scale)) for models M1, M2, and M3 with different porosities *P* and different passivation types. The black crosses (+) and exes (x) show the experimental data for the as-fabricated and passivated samples, respectively [21]; orange squares show the MD simulation result for the case without coating; green circles show the MD simulation with the –O coating; blue up-pointing triangles show the MD simulation with the –H coating; and red down-pointing triangles show the MD simulation with the –OH coating.

**Table 1 materials-15-08678-t001:** Areas and percentage of the surface atoms for models M1, M2, and M3.

	Porosity (%)	Internal Surface Area ($\frac{m^{2}}{g}$)	$\frac{N_{S}}{N}$ (%)
M1 (aero)	95	100.7	20.2
M2 (round)	50	119.6	27.5
M2 (round)	63	151.0	37.04

**Table 2 materials-15-08678-t002:** Heat capacity $C_{P}$ ($\frac{J}{kgK}$) obtained by the direct MD calculation (Equation 7) and the calculation through the PDOS (Equation 9) at a temperature of 300 K.

	**Direct MD Calculation**			**Calculation from PDOS**		
bulk Si	839			855		
	**M1**	**M2**	**M3**	**M1**	**M2**	**M3**
S **(m^2^/g)**	100.7	119.6	151.0	100.7	119.6	151.0
Uncoated	853	878	905	869	878	901
–O coating	950	988	1079	1020	1026	1023
–H coating	968	1028	1139	991	1135	1225
–OH coating	1086	1151	1227	1116	1162	1224

**Table 3 materials-15-08678-t003:** Thermal conductivity κ in $\left(\frac{W}{mK}\right)$ obtained by the MD calculation at a temperature of 300 K and experimental data [21] for models M1, M2, and M3 with different porosities *P* and inner surface passivation types. The values in brackets are $k^{\prime }$.

		Experimental Data [21]		MD Calculation			
	*P*, (%)	As- Fabricated	Passivated	no	–O	–H	–OH
Bulk	0	149	-	175	-	-	-
M1	95	<0.13	<0.666	0.159	0.118	0.126	0.262
		(3.98)	(5.67)	(7.94)	(5.90)	(6.30)	(13.1)
M2	50	1.45	2.48	3.75	1.17	3.00	7.34
		(3.98)	(5.67)	(7.82)	(2.44)	(6.25)	(15.2)
M3	63	0.787	1.65	0.86	0.46	0.75	2.29
		(3.98)	(5.67)	(2.53)	(1.35)	(2.21)	(8.52)

**Table 4 materials-15-08678-t004:** Thermal effusivity ϵ in $\left(\frac{J}{m^{2}K\sqrt{s}}\right)$ and thermal diffusivity α in ($\frac{mm^{2}}{c}$) for modeled bulk Si and M1, M2, and M3 and internal surface passivation types at a temperature of 300 K.

	**Thermal Effusivity**			**Thermal Diffusivity**		
bulk Si	18,490			89.52		
	**M1**	**M2**	**M3**	**M1**	**M2**	**M3**
Uncovered	128.9	1990	853.6	1.521	3.552	1.015
–O cover	123.7	1227	743.6	0.910	0.910	0.383
–H cover	122.6	1935	895.9	1.055	2.404	0.701
–OH cover	197.5	3447	1779	1.759	4.534	1.657

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
