# Peer review of "Thermal Properties of Porous Silicon Nanomaterials"

_materials, 2022, doi:10.3390/ma15238678_

Round 1
Reviewer 1 Report
The authors have theoretically investigated the thermal properties of porous silicon. Numerous works have be previously reported on evaluating the thermal properties of silicon-based on porosity. However, the current work seems interesting but lacks at various points:
1) Abstract: Kindly make the abstract more quantitative by adding the thermal properties such as thermal conductivity, effusivity, and diffusivity of porous silicon.
2) Introduction: The introduction lacks to provide the recent literature. Silicon plays a vital role in numerous energy conversion and storage applications due to its narrow bandgap and semiconductor properties. Authors need to add recent literature on different silicon nanostructures and their properties with future applications.
Some of the suggestions are:
1) Khanna S, Marathey P, Paneliya S, Vanpariya A, Ray A, Banerjee R, Mukhopadhyay I. Fabrication of silicon nanohorns via soft lithography technique for photoelectrochemical application. International Journal of Hydrogen Energy. 2021 Apr 29;46(30):16404-13.
2) Tsamis C, Nassiopoulou AG, Tserepi A. Thermal properties of suspended porous silicon micro-hotplates for sensor applications. Sensors and Actuators B: Chemical. 2003 Oct 15;95(1-3):78-82.
Even silicon oxide nanomaterials have a high heat capacity and have been used as core-shell nanostructures for TES applications.
3) Khanna S, Paneliya S, Prajapati P, Mukhopadhyay I, Jouhara H. Ultra-stable silica/exfoliated graphite encapsulated n-hexacosane phase change nanocomposite: A promising material for thermal energy storage applications. Energy. 2022 Jul 1;250:123729.
Result and Discussion:
Authors are requested to add tables for determining the thermal properties such as thermal conductivity, effusivity, and diffusivity for the porous silicon material.
The manuscript lacks the novelty part; thus, a comparative study with the reported work is required to show the novelty of the work.
Kindly add some reported experimental work that these theoretical works support.
Do the thermal properties change drastically with a change in porosity? Explain
Conclusion: The section needs to be revised more quantitatively.
There are some grammatical errors that need to be corrected in the revised version of the manuscript.
Author Response
Dear Reviewer 1:
We appreciate the thorough and constructive report of knowledgeable reviewers. We are greatly encouraged by the reviewers opinion that our work is suitable for publication in the Materials.
We have addressed all points raised by the reviewers in the response written below. The appropriate changes are highlighted by red in the text of the revised manuscript.
Reviewer N1 Comments:
Q1. Abstract: Kindly make the abstract more quantitative by adding the thermal properties such as thermal conductivity, effusivity, and diffusivity of porous silicon.
A1. According to the referee’s remark we have include the paragraph concerning quantitative comparison of thermal properties of nanoporous silicon in comparison with bulk material.
Original text (1 page):
“It has been shown also the thermal conductivity of the investigated materials depends on the degree of their porosity and can be changed significantly upon passivation of their inner surfaceb
Revised text:
“It has been shown also that the thermal conductivity of the investigated materials depends on the degree of their porosity and can be changed significantly upon passivation of their inner surface with different ions.
\textcolor{red}{We have demonstrated that in the various simulated types of porous silicon, the thermal conductivity changes by 1–2 orders in comparison to the bulk silicon. At the same time, it was noted that the nature of the passivation of the internal surfaces of nanosilicon affects to the thermal conductivity. For example, passivation of surfaces with hydrogen does not change it significantly, but passivation with oxygen ions reduces thatby an average of 2 times and passivation with hydroxyl ions increases the thermal conductivity by 2-3 times. During passivation the same trends are observed for the change in thermal effusivities and diffusivities for all cases of nanoporous silicon, but the changes were smaller (1.5–2 times).
In conclusion, the work shows ways to change the thermal properties of new nanostructured materials, which is important for their practical applications.}
Q2. Introduction: The introduction lacks to provide the recent literature. Silicon plays a vital role in numerous energy conversion and storage applications due to its narrow bandgap and semiconductor properties. Authors need to add recent literature on different silicon nanostructures and their properties with future applications.
Some of the suggestions are:
1) Khanna S, Marathey P, Paneliya S, Vanpariya A, Ray A, Banerjee R, Mukhopadhyay I. Fabrication of silicon nanohorns via soft lithography technique for photoelectrochemical application. International Journal of Hydrogen Energy. 2021 Apr 29;46(30):16404-13.
2) Tsamis C, Nassiopoulou AG, Tserepi A. Thermal properties of suspended porous silicon micro-hotplates for sensor applications. Sensors and Actuators B: Chemical. 2003 Oct 15;95(1-3):78-82.
Even silicon oxide nanomaterials have a high heat capacity and have been used as core-shell nanostructures for TES applications.
3) Khanna S, Paneliya S, Prajapati P, Mukhopadhyay I, Jouhara H. Ultra-stable silica/exfoliated graphite encapsulated n-hexacosane phase change nanocomposite: A promising material for thermal energy storage applications. Energy. 2022 Jul 1;250:123729.
A2. We are grateful to the reviewer for mentioning these valuable articles. We have included them in the article references. We also additionally have found works concerning using of nanostructured silicon for various energy storage applications and updated the introduction:
Revised text
page 1:
\textcolor{red}{Also it was shown that porous silicon could be used for biodetection of miRNA based on surface plasmon resonance and surface enhanced Raman scattering (SERS)\cite{vendamani2022silicon}.}
page 2:
\textcolor{red}{In addition to porous silicon, other silicon nanostructures can be used for these purposes, for example, nanoparticle \cite{shin2020sustainable} or nanowires \cite{bogart2014lithium}.}
\textcolor{red}{Porous silicon also find application as a wide-gap absorber for solar cells. Its advantages consist of increased explosion to illumination due to presence of surface area and possibility of the conversion of high-energy ultraviolet and blue part of the solar spectrum \cite{dzhafarov2018porous}. }
\textcolor{red}{ As for different silicon nanostructures, there are currently many ways how it allows for improved energy storage methods. For example, a recent study showed how silicon nanohorns can be used to enhance the performance of solar water splitting related devices that provide better energy conversion as compared to their bulk counterparts \cite{khanna2021fabrication}. Another study show that large-area ordered Si nanowire arrays \cite{huang2014large} can be used for the same purpose.}
\textcolor{red}{ Silicon nanowires, porous membranes and nanomeshes could convert heat flows directly into useable electrical energy that allow application of such material as a thermoelectric generator\cite{schierning2014silicon}.}
\textcolor{red}{Futher it was shown that porous silicon micro-hotplates could provide improved thermal isolation in silicon thermal sensor devices because it can be easily obtained compatible with standard Complementary metal–oxide–semiconductor (CMOS) technology \cite{tsamis2003thermal}.}
\textcolor{red}{It also important to mention, that silicon oxide nanomaterials have a high heat capacity and have been used as core-shell nanostructures for TES applications \cite{khanna2022ultra}.}
Q3. Result and Discussion:
Authors are requested to add tables for determining the thermal properties such as thermal conductivity, effusivity, and diffusivity for the porous silicon material.
A3.
Thank you for this valuable remark regarding the mention of the thermal effusivity and diffusivity. We have included a paragraph regarding their definition and use. Also we have added Table 4 with their calculated values
Revised text (page 12):
\textcolor{red}{
In addition to thermal conductivity, important quantities in understanding the thermal processes inside a material are thermal effusivity and thermal diffusivity which calculated data are presented in Table \ref{effusivity_table}. Thermal effusivity or thermal responsivity of material is obtained by $\epsilon= \sqrt{\kappa \rho C_P }$. It helps better understand the porous silicon ability to exchange thermal energy with surroundings. Thermal diffusivity is determined as $\alpha= \frac{\kappa}{\rho C_P}$. It measures the rate of transfer of heat of a material from the hot end to the cold end.}
Q4. The manuscript lacks the novelty part; thus, a comparative study with the reported work is required to show the novelty of the work. Kindly add some reported experimental work that these theoretical works support.
A4.
The novelty of our work that is the first theoretical study of the porous silicon thermal properties by MD method. As we mentioned in the Abstract, we have defined the thermal conductivities decreased by 1–2 orders in comparison with the bulk silicon. We have compared the calculated thermal conductivity and experimental study. Our study have showed also the thermal conductivity depend on which passivating ions are on the silicon nanomaterials internal surface.
Revised text (page 11):
“\textcolor{red}{ Our calculated porous silicon thermal conductiviies agrees with available experimental data \cite{erfantalab2022determination}. Our study also showed the result of thermal conductivity depending on which passivating ion is on the surface.}”
We also have compared in more detail the our new computed and experimental heat capacities values.
Revised text (page 8-9):
\textcolor{red}
{In the experimental study \cite{erfantalab2022determination} it was found that the specific heat of as-fabricated porous silicon samples varies from 0.8 to 2.1 ($\frac{J}{kgK}$) when the porosity varies from 45\% to 77\%. Unfortunately, the pore size has not been determined. It was only determined that the size ranged from 4 to 70 nm. The internal surface area has not been measured or estimated also. Our MD simulations show the heat capacity depend on the internal surface area, see Fig.\ref{c_pic}. One can see the heat capacity grows monotonically with the growth of the Si nanostructures inner surface, and this growth reaches ~ 30\%. This dependence can be explained from a detailed consideration of the formulas (\ref{CPO_eq}) and (\ref{CPA_eq}), where the low-frequency contribution to the density of phonon states increase the heat capacity.} \textcolor{red}{Also, in contrast to the experimental studies, the influence of different passivated ions can be considered in more detail in our MD simulation.}
Q5. Do the thermal properties change drastically with a change in porosity? Explain
A5.
Yes. We show that the heat capacity (Fig.3) and especially the thermal conductivity (Fig.5 and Table 3) change quite significantly. For example, the thermal conductivity decreased by 1–2 orders when porosity increase. We summarize these changes in Abstract and Conclusions.
Q6. Conclusion: The section needs to be revised more quantitatively.
A6.
Thank you for this note. we finalized the text of the conclusion and made it more specific and quantitative.
Revised text (page 12):
\textcolor{red}{We have demonstrated that in the various simulated types of porous silicon, the thermal conductivity changes by 1–2 orders in comparison to the bulk silicon. At the same time, it was noted that the nature of the passivation of the internal surfaces of nanosilicon affects to the thermal conductivity. For example, passivation of surfaces with hydrogen does not change it significantly, but passivation with oxygen ions reduces thatby an average of 2 times and passivation with hydroxyl ions increases the thermal conductivity by 2-3 times. During passivation the same trends are observed for the change in thermal effusivities and diffusivities for all cases of nanoporous silicon, but the changes were smaller (1.5–2 times).}
Q7. There are some grammatical errors that need to be corrected in the revised version of the manuscript.
A7. We have corrected the text
In addition, in accordance with the wishes of the reviewer 1 and reviewer 2 we carefully checked the text of the manuscript, made grammatical corrections, and improved the language.
Reviewer 2 Report
The heat capacity, thermal conductivity, and phonon density of states of silicon-based nanomaterials was analyzed by the molecular dynamics calculation in this work. The results of this study can be used to develop the thermal characteristics of silicon-based nanomaterials.
The influence of different passivation groups on the thermal properties of materials has been explored here, while what effect does the content of etching groups have on the relevant properties?
Author Response
Dear Reviewer2:
We appreciate the thorough and constructive report of knowledgeable reviewers. We are greatly encouraged by the reviewers opinion that our work is suitable for publication in the Materials.
We have addressed all points raised by the reviewers in the response written below. The appropriate changes are highlighted by red in the text of the revised manuscript.
Q1. The influence of different passivation groups on the thermal properties of materials has been explored here, while what effect does the content of etching groups have on the relevant properties?
A1. We thank the reviewer for a useful remark. In fact, in experimental samples, etching groups can affect the ongoing chemical reactions. However, in our study, we did not model the surface chemistry in the presence of etching groups.
Revised text (page 9):
“...Also we have not modelled the etching group which correspond to the experimental manufacturing process and which take part in reactions...”
In addition, in accordance with the wishes of the reviewer 1 and reviewer 2 we carefully checked the text of the manuscript, made grammatical corrections, and improved the language.
Finally, we would like to thank reviewers for the useful remarks which allow us to make our paper better and clearer for the reader.
Reviewer 3 Report
I am reviewing by Thermal properties of porous silicon nanomaterials
The authors present porous silicon thermal properties determination by using Molecular Dynamics, where the main finding is related to the high heat capacity and low thermal conductivity of porous silicon. The subject is interesting and worth studying. However, the article, at present, is not suitable for publication in Materials. Therefore, some aspects must be improved and/or revised. Below I summarize some minor comments and the main observation related to the experimental validation of the significant findings.
1. The critical issue of porous silicon is the manufacturing process because the electrochemical reaction does not occur under a potenciostatic regime. Alternatively, some works have reported in situ techniques, such as optical interferometry, photoacoustics, and impedancymetry, to monitor de Porous silicon formation. The author should mention some of these experimental works because it is the way to control the porosity and can be correlated with Molecular Dynamics simulations.
2. In the sentence (line 42), The monotonic growth of the heat capacity was first established as a function of the degree of porosity. Please, a careful revision of the cited paper to include the porosity interval for the heat capacity has a growing tendency. Also, indicate if the porous were filled with air or any residue of the photolithography process. This is important to rule out any interaction of the porous structure with the filling.
3. The surface chemistry of porous silicon is varied. It is H-terminated in O and F-terminated just after fabrication by electronic anodizing in HF-based electrolytes. However, these chemical species are not stable and tend to be reabsorbed. Some works are related to the chemical evolution of the surface as a function of temperature and how the surface of porous silicon tends to oxidize because of temperature. The authors should make more references to experimental studies to validate the simulation of MD.
4. At what temperature is the system?
5. Chemical species, especially with H-terminations, easily desorb and are replaced by oxygen (authors should review recent publications showing this desorption via FTIR). In what temperature ranges is the proposed simulation valid?
Are you considering the possible reaction of desorption and subsequent oxidation?
Author Response
Dear Reviewer 3:
We appreciate the thorough and constructive report of knowledgeable reviewers. We are greatly encouraged by the reviewers opinion that our work is suitable for publication in the Materials.
We have addressed all points raised by the reviewers in the response written below. The appropriate changes are highlighted by red in the text of the revised manuscript.
Q1. The critical issue of porous silicon is the manufacturing process because the electrochemical reaction does not occur under a potenciostatic regime. Alternatively, some works have reported in situ techniques, such as optical interferometry, photoacoustics, and impedancymetry, to monitor de Porous silicon formation. The author should mention some of these experimental works because it is the way to control the porosity and can be correlated with Molecular Dynamics simulations.
A1.
We thank the reviewer for useful additions. Indeed, the added paragraph sheds light on the fact that porous structures can be varied due to existing control methods.
Revised text (page 4):
\textcolor{red}{
In practice there are different ways to control porosity, pore size and internal structures, which allow to obtain a material with desired properties. For that can be used an optical interferometry \cite{ramirez2018optical}, photoacoustics \cite{ramirez2018optical}, and impedancymetry \cite{husairi2014electrochemical}techniques. It is already described cases of pore types being blind, interconnected, completely isolated or through pores. Also it is possible to obtain different shape of the pores for example cylindrical, inkwell, funnel, cuboid, triangular or pyramidal and so on..}
Q2. In the sentence (line 42), The monotonic growth of the heat capacity was first established as a function of the degree of porosity. Please, a careful revision of the cited paper to include the porosity interval for the heat capacity has a growing tendency. Also, indicate if the porous were filled with air or any residue of the photolithography process. This is important to rule out any interaction of the porous structure with the filling.
A2.
According to the comment of the reviewer, the following paragraph was added to the article, which more accurately explains the experimental study cited by us.
Revised text (page 8):
\textcolor{red}
{In the experimental study \cite{erfantalab2022determination} it was found that the specific heat of as-fabricated porous silicon samples varies from 0.8 to 2.1 ($\frac{J}{kgK}$) when the porosity varies from 45\% to 77\%. Unfortunately, the pore size has not been determined. It was only determined that the size ranged from 4 to 70 nm. The internal surface area has not been measured or estimated also. Our MD simulations show the heat capacity depend on the internal surface area, see Fig.\ref{c_pic}.One can see the heat capacity grows monotonically with the growth of the Si nanostructures inner surface, and this growth reaches ~ 30\%, see Fig.\ref{c_pic}. This dependence can be explained from a detailed consideration of the formulas (\ref{CPO_eq}) and (\ref{CPA_eq}), where the low-frequency contribution to the density of phonon states increase the heat capacity.} \textcolor{red}{Also, in contrast to the experimental studies, the influence of different passivated ions can be considered in more detail in our MD simulation.}
It is also important to note that the simulated porous silicon is not filled with air or any residue of the photolithography process, therefore, the following sentence has been added to clarify this.
Revised text (page 9):
“Also we have not modelled the etching group or air which correspond to the experimental manufacturing process and which take part in reactions.”
Q3. The surface chemistry of porous silicon is varied. It is H-terminated in O and F-terminated just after fabrication by electronic anodizing in HF-based electrolytes. However, these chemical species are not stable and tend to be reabsorbed. Some works are related to the chemical evolution of the surface as a function of temperature and how the surface of porous silicon tends to oxidize because of temperature. The authors should make more references to experimental studies to validate the simulation of MD.
A3. We thank the reviewer for a kind remark about surface evolution in porous silicon. As suggested by the reviewer, we paid attention to the process of surface evolution and cited two papers that show surface aging and oxidation.
Revised text (page 9):
\textcolor{red}
{It is important to mention that the porous silicon surface tend to chemical evolution in experiments. For example, the hydrogen desorption reactions were observed in the experimental samples of porous silicon \cite{rivolo2003joint}, which affected to the heat capacity and thermal conductivity values. The process of oxidation was observed in the work \cite{loni2013exothermic} which lasted about an hour. In \cite{noval2012aging} the process of aging in aquas solutions showed changing of surface properties what lasts for several days. }
In our MD simulations the reactions of desorption and reabsortion of three different ions were not occurred during total MD simulation. Also we have not modelled the etching group or air which correspond to the experimental manufacturing process and which take part in reactions. Besides, our MD simulation lasts (~1nsec.), so it is not allowed to track the influence of relatively slow surface evolution processes.}
Q4. At what temperature is the system?
A4 In the section Molecular dynamic calculations it was mentioned that all calculations are carried out at a temperature of 300K, however, the reviewer correctly noted that this was not mentioned in the section in the discussion. Therefore, an explanation was added to the description of Table 2, Table 3 and Table 4 that these data are shown for a temperature of 300K
Q5. Chemical species, especially with (authors should review recent publications showing this desorption via FTIR). In what temperature ranges is the proposed simulation valid?
A5 All MD simulations were carried out at 300K so presented data are relevant for this temperature. We have already answered in the answer A3 that the desorption reactions were not observed during MD simulations, but they were observed in the long-term experiments.
Q6 Are you considering the possible reaction of desorption and subsequent oxidation?
A6 It is known that reaction of desorption and subsequent oxidation are possible in experiments, see Answer A3 again. One of the main goals of our work was to trace the effect of adsorbed ions on the thermal properties of the considered nanomaterials.
In addition, in accordance with the wishes of the reviewer 1 and reviewer 2 we carefully checked the text of the manuscript, made grammatical corrections, and improved the language.
Finally, we would like to thank reviewers for the useful remarks which allow us to make our paper better and clearer for the reader.
Round 2
Reviewer 1 Report
The authors have significantly improved the manuscript; thus, it can be accepted in its present form.